

# Global Analysis of SSAs and the Impact of the EIC and SoLID on Tensor Charge Extractions

**Daniel Pitonyak⋆**

Department of Physics, Lebanon Valley College, Annville, PA 17003, USA

⋆ pitonyak@lvc.edu

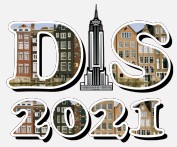

*Proceedings for the XXVIII International Workshop
on Deep-Inelastic Scattering and Related Subjects,
Stony Brook University, New York, USA, 12-16 April 2021*

## Abstract

Single transverse-spin asymmetries (SSAs) give great insight into the 3-dimensional structure of hadrons. We report on the first global QCD analysis of SSAs in semi-inclusive deep-inelastic scattering, electron-positron annihilation, Drell-Yan, and single-inclusive proton-proton collisions. One byproduct of the analysis is an extraction of the transversity function, from which the nucleon tensor charges can be computed, and we find, for the first time, agreement with lattice QCD for these quantities. Based on this analysis, we perform an impact study of future data on extractions of the nucleon tensor charges.

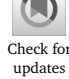

## 1 Introduction

The 3-dimensional structure of hadrons has been an intense area of research for over two decades. Central to these investigations are single transverse-spin asymmetries (SSAs), which arise in reactions where a hadron or quark is transversely polarized. In processes sensitive to the intrinsic motion of partons, like semi-inclusive deep-inelastic scattering (SIDIS) ($\ell N \to \ell h X$), semi-inclusive electron-positron annihilation (SIA) ($e^+ e^- \to h_1 h_2 X$), and Drell-Yan lepton pair or weak gauge boson production (DY) ($pp \to \{\ell^+ \ell^-, W^\pm, Z\} X$), SSAs are analyzed within transverse momentum dependent (TMD) factorization. Several such observables of note are the Sivers effect and Collins effect in SIDIS, Collins effect in SIA, and Sivers effect in DY. They allow one to probe several TMD parton distribution functions (PDFs) and fragmentation functions (FFs), namely, the Sivers function $f_{1T}^\perp(x, k_T)$, Collins function $H_1^\perp(z, p_T)$, and transversity function $h_1(x, k_T)$, where the momentum fractions and transverse momenta are indicated by the arguments.

On the other hand, there are certain observables that are sensitive only to the collinear motion of partons that nevertheless provide complementary information to TMD observables. The main such reaction is $A_N$ in single-inclusive proton-proton collisions ($pp \to h X$), which is

sensitive to collinear twist-3 multi-parton correlators, namely $F_{FT}(x,x), H_1^{\perp(1)}(z)$, and $\tilde{H}(z)$. The dominant source of $A_N$ is due to a coupling between the (collinear) transversity function $h_1(x)$ and $H_1^{\perp(1)}(z)$ [1–3]. Moreover, in a simple parton model picture, some of these collinear twist-3 functions can be related to transverse-momentum moments of the aforementioned TMD functions [4]. Furthermore, the Operator Product Expansion (OPE) of TMDs [5–8] in the Fourier-conjugate position space also can be written, e.g., in the case of the Sivers and Collins functions, in terms multi-parton correlators, specifically $\tilde{f}_{1T}^{\perp}(x,b_T) \xrightarrow{\text{small } b_T} F_{FT}(x,x)$ and $\tilde{H}_1^{\perp}(z,b_T) \xrightarrow{\text{small } b_T} H_1^{\perp(1)}(z)$. (Note that one also has $\tilde{h}_1(x,b_T) \xrightarrow{\text{small } b_T} h_1(x)$.) Ultimately, these dynamical twist-3 functions are what underlie all SSAs. Recently, we demonstrated this common origin of SSAs in a phenomenological analysis [3]. In Sec. 2 we review the findings of that work. One main outcome was a computation of the nucleon tensor charges $\delta q \equiv \int_0^1 dx[h_1^q(x) - h_1^{\bar{q}}(x)]$ and $g_T \equiv \delta u - \delta d$, which, for the first time, agreed with lattice QCD calculations. In Sec. 3 we discuss our study in Ref. [9] of the impact future data from the Electron-Ion Collider (EIC) [10], to be built at Brookhaven National Lab, and SoLID at Jefferson Lab [11, 12], will have on tensor charge extractions. We summarize our main results in Sec. 4.

## 2 Global Analysis of SSAs

In Ref. [3] we performed the first global QCD analysis of the SSAs listed in Table 1 using a Gaussian ansatz: $F(x,k_T) \sim F(x)e^{-k_T^2/\langle k_T^2 \rangle}$ for a generic TMD. The $x$ ($z$) dependence had the following functional form:

$$F^q(x) = \frac{N_q\, x^{a_q}(1-x)^{b_q}(1+\gamma_q\, x^{\alpha_q}(1-x)^{\beta_q})}{\mathrm{B}[a_q+2, b_q+1] + \gamma_q \mathrm{B}[a_q+\alpha_q+2, b_q+\beta_q+1]}, \tag{1}$$

where $F^q = h_1^q, \pi F_{FT}^q, H_{1h/q}^{\perp(1)}$ (with $x \to z$ for the Collins function), and $B$ is the Euler beta function. We used a Monte Carlo framework to reliably sample the Bayesian posterior distribution for the parameters. One can see in Table 1 that we achieve a very good description of all data sets. The non-perturbative functions inferred from our analysis are shown in the left panel of Fig. 1.

In the right panel of Fig. 1 we display our computed nucleon tensor charges using the transversity function $h_1(x)$. For the first time there is agreement between phenomenology and lattice QCD for these quantities. This highlights the importance of performing a simultaneous analysis of all SSA observables, in particular including $A_N$. Notice that the green ellipse, from a fit of only SIDIS and SIA data, does not match with lattice. Once the additional data from $A_N$ is included, the Global result in red shifts over closer to the lattice points. That is, additional data sets (in this case $A_N$) can cause the non-perturbative functions to "shuffle" around to another solution that still gives good agreement with the SIDIS and SIA observables. Furthermore, the inclusion of $A_N$ allowed us to achieve the most precise phenomenological determination of the nucleon tensor charges to date.

The results of our combined analysis indicate SSAs have a common origin. Namely, they are due to the intrinsic quantum-mechanical interference from multi-parton states. Our findings imply that the effects are predominantly non-perturbative and intrinsic to hadronic wavefunctions. Future data, like those from JLab-12 GeV, COMPASS, an upgraded RHIC, Belle II, and the EIC will help to reduce the uncertainties of the extracted functions. We discuss the impact of two future data sets on the extraction of the nucleon tensor charges in the next section.

Table 1: Summary of the SSAs analyzed in our global fit and the $\chi^2/N_{\text{pts.}}$ obtained for each observable.

| Observable | Reactions | $\chi^2/N_{\text{pts.}}$ |
|---|---|---|
| $A_{\text{SIDIS}}^{\text{Siv}}$ | $e + (p, d)^{\uparrow} \rightarrow e + (\pi^+, \pi^-, \pi^0) + X$ | $150.0/126 = 1.19$ |
| $A_{\text{SIDIS}}^{\text{Col}}$ | $e + (p, d)^{\uparrow} \rightarrow e + (\pi^+, \pi^-, \pi^0) + X$ | $111.3/126 = 0.88$ |
| $A_{\text{SIA}}^{\text{Col}}$ | $e^+ + e^- \rightarrow \pi^+ \pi^- (UC, UL) + X$ | $154.5/176 = 0.88$ |
| $A_{\text{DY}}^{\text{Siv}}$ | $\pi^- + p^{\uparrow} \rightarrow \mu^+ \mu^- + X$ | $5.96/12 = 0.50$ |
| $A_{\text{DY}}^{\text{Siv}}$ | $p^{\uparrow} + p \rightarrow (W^+, W^-, Z) + X$ | $31.8/17 = 1.87$ |
| $A_N^h$ | $p^{\uparrow} + p \rightarrow (\pi^+, \pi^-, \pi^0) + X$ | $66.5/60 = 1.11$ |

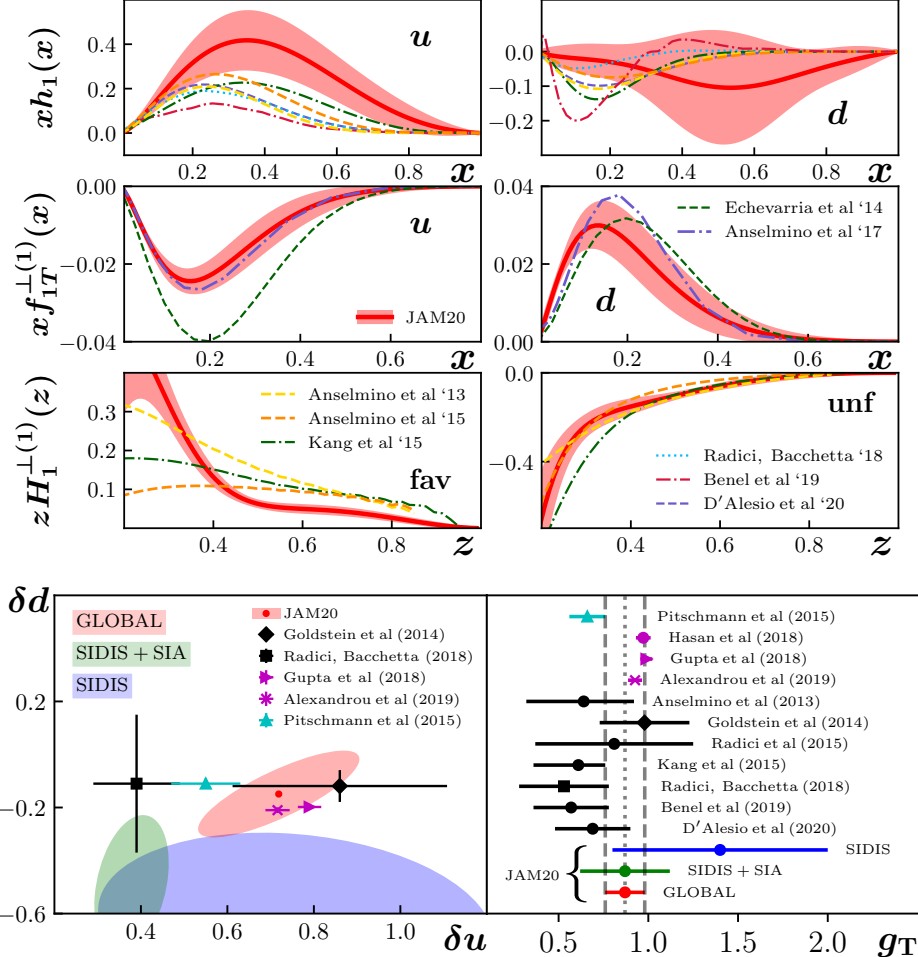

Figure 1: (Top) The extracted functions $h_1(x)$, $f_{1T}^{\perp(1)}(x)$, and $H_1^{\perp(1)}(z)$ at $Q^2 = 4\,\text{GeV}^2$ from our (JAM20) global analysis (red solid curves with 1-$\sigma$ CL error bands). The functions from other groups [13–20] are also shown. (Bottom) The tensor charges $\delta u$, $\delta d$, and $g_T$. Our (JAM20) results at $Q^2 = 4\,\text{GeV}^2$ along with others from phenomenology (black) [13, 15, 18–22], lattice QCD (purple) [23–25], and Dyson-Schwinger (cyan) [26].

## 3 Impact Study of Future Data from EIC and SoLID

The nucleon tensor charges in the right panel of Fig. 1 from our global analysis still have relatively large uncertainties compared to lattice QCD calculations. Therefore, in Ref. [9] we assessed the impact of future data from the EIC and SoLID at Jefferson Lab. The EIC will provide a broad kinematic coverage in $x$ and $Q^2$, spanning several decades in both quantities. SoLID will be able to probe larger $x$ and lower $Q^2$ at a higher luminosity than the EIC. The kinematic coverage and uncertainties expected from both experiments is shown in Fig. 2.

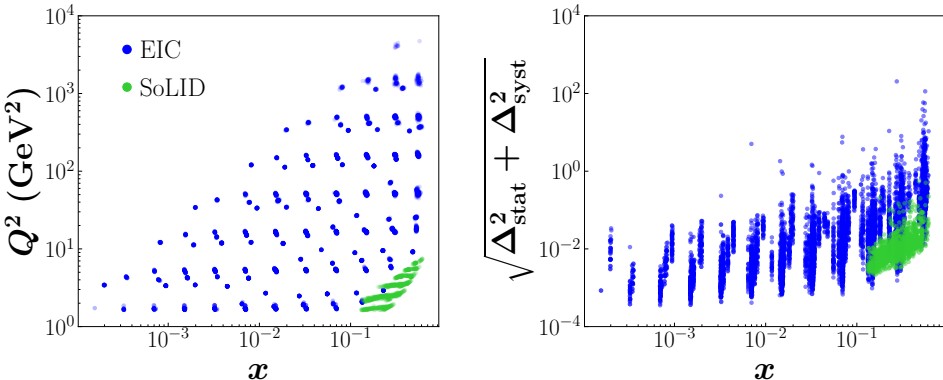

Figure 2: (Left) Scatter plot of the $x$ and $Q^2$ coverage of the EIC (blue points) and SoLID (green points). (Right) The quadrature of statistical ($\Delta_{\text{stat}}$) and systematic ($\Delta_{\text{syst}}$) errors of the pseudo-data for the Collins effect in SIDIS plotted versus $x$.

We generated pseudo-data for the Collins effect in SIDIS for both the EIC and SoLID for the various anticipated center-of-mass energies and hadronic configurations. Using our global analysis in Sec. 2 (JAM20) as a baseline, we then were able to determine the impact of each facility on the extraction of the transversity and Collins functions and, subsequently, the nucleon tensor charges. We first focus on the EIC. In Fig. 3 we show the results for $h_1(x)$ and $H_1^{\perp(1)}(z)$ as well as $\delta u, \delta d$, and $g_T$. In particular, we highlight that the $^3He$ program at the EIC is crucial to obtain a significant reduction in the down quark transversity function and de-correlate the extraction of $\delta u$ and $\delta d$. In the end, we expect that EIC data will allow phenomenological extractions of the nucleon tensor charges to be as precise as current lattice calculations.

Turning now to SoLID, we show in the top panel of Fig. 4 the relative uncertainty of $h_1(x)$. Notice that for the down quark, SoLID, which has better capabilities with $^3He$ than the EIC, will show a greater reduction at larger $x$. Ultimately, combining both data sets gives the greatest reduction in the uncertainties. Similarly, as seen in the bottom panel of Fig. 4, SoLID itself will be able to significantly reduce the uncertainties in the nucleon tensor charges close to the level of lattice QCD. In the end, the EIC+SoLID fit gives the most precise extraction.

We emphasize that a precise measurement cannot always guarantee a very accurate extraction of PDFs and FFs, and multiple experiments, such as EIC and SoLID, should be performed in a wide kinematical region in order to minimize bias and expose any potential tensions between data sets. Given that the tensor charge is a fundamental charge of the nucleon and connected to searches for BSM physics [27–29], future precision measurements from the EIC and Jefferson Lab sensitive to transversity are of utmost importance and necessary to see if a consistent picture emerges for the value of the tensor charge of the nucleon.

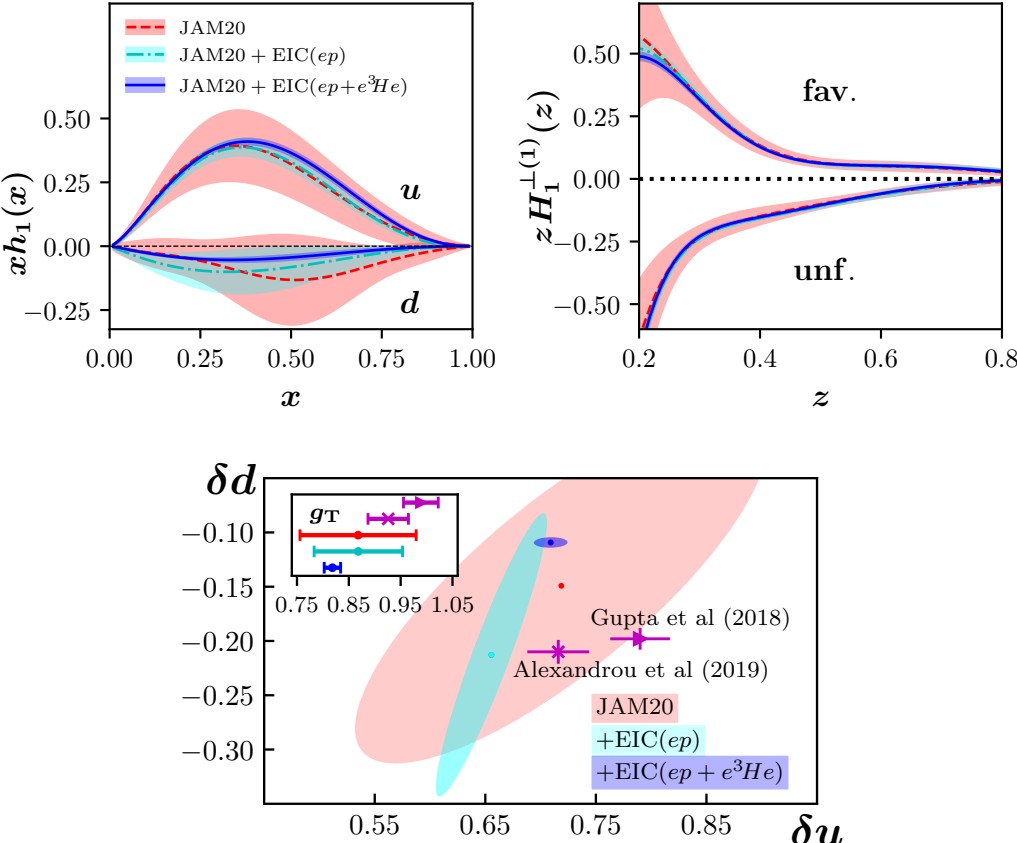

Figure 3: (Top) Plot of the transversity function for up and down quarks as well as the favored and unfavored Collins function first moment from the JAM20 global analysis (light red band with the dashed red line for the central value) as well as a re-fit that includes EIC Collins effect pion production pseudo-data for a proton beam only (cyan band with the dot-dashed cyan line for the central value) and for both proton and $^3He$ beams together (blue band with the solid blue line for the central value). (Bottom) Individual flavor tensor charges $\delta u$, $\delta d$ as well as the isovector charge $g_T$ for the same scenarios. Also shown are the results from two recent lattice QCD calculations (purple) [23, 25].

## 4  Conclusion

We have performed the first global analysis of SSAs in SIDIS, DY, $e^+e^-$ annihilation (SIA), and proton-proton ($A_N$) collisions and extracted a universal set of non-perturbative functions, showing a common origin of SSAs. First agreement with lattice QCD on the tensor charges of the nucleon was obtained, but still with large uncertainties. EIC data on the Collins effect will allow for phenomenological extractions of the nucleon tensor charges to be as precise as current lattice calculations. SoLID at JLab will also provide important constraints. In order to reduce bias and obtain the most accurate extraction of $\delta u, \delta d$, and $g_T$, one must have data from multiple future facilities that give the most kinematic coverage possible in $x$ and $Q^2$.

**Funding information**  This work has been supported by the National Science Foundation under Grant No. PHY-2011763 (D.P.) and within the framework of the TMD Topical Collaboration.

![Sci|Post] 

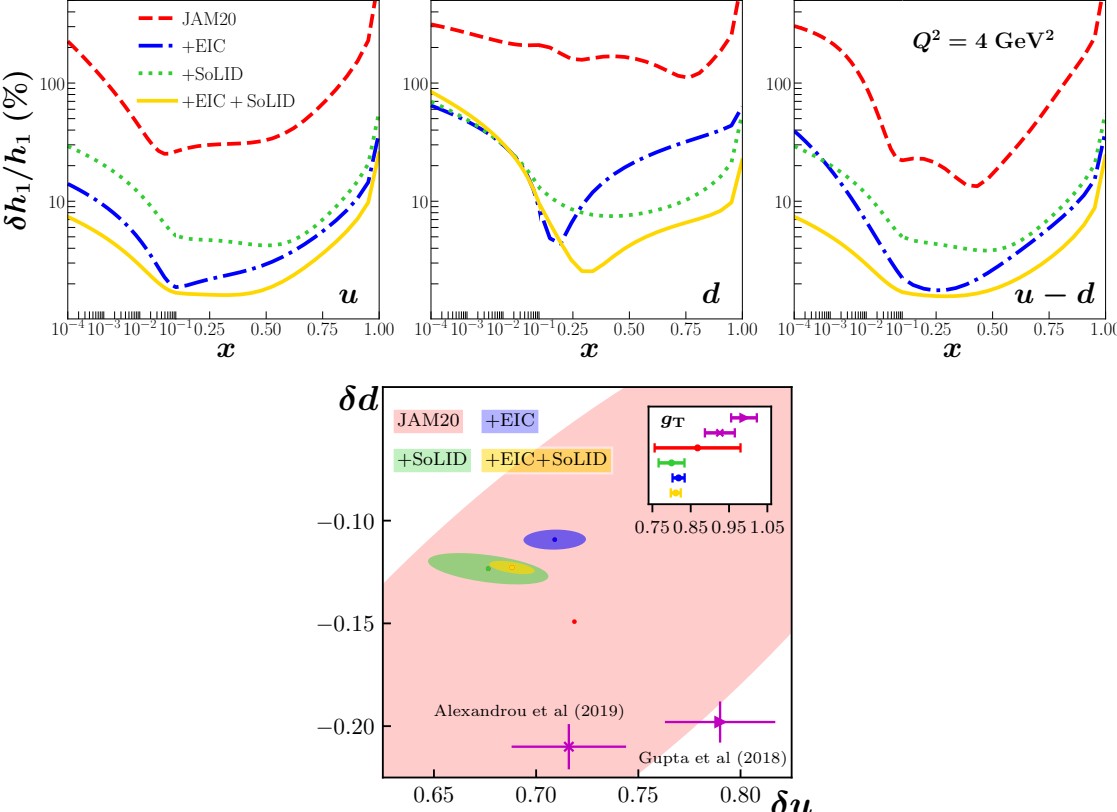

Figure 4: (Top) The ratio of the error of transversity to its central value for $u$, $d$, and $u-d$ for JAM20 (red dashed line), JAM20+EIC pseudo-data (blue dash-dotted line), JAM20+SoLID pseudo-data (green dotted line), and JAM20+EIC+SoLID pseudo-data (gold solid line). (Bottom) Individual flavor tensor charges $\delta u$, $\delta d$ as well as the isovector charge $g_T$ for the same scenarios.

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
