# Peer review of "Global Analysis of SSAs and the Impact of the EIC and SoLID on Tensor Charge Extractions"

_SciPost Physics Proceedings, doi:SciPost Phys. Proc. 8, 044 (2022)_

## Round 1 · Referee Report · Anonymous (Referee 1) · 2022-2-28

Report

The paper, "Global Analysis of SSAs and the Impact of the EIC and SoLID on Tensor Charge Extractions", reports on the first global QCD analysis of SIDIS, SIA, DY, and single inclusive proton-proton scattering. In this analysis nucleon tensor charges are computed and found to be in agreement with lattice calculations for the first time. The impact of future measurements on the extracted values is also performed.

The results shown in this work are compelling and well described. The summary of the analyzed SSAs are concise and appear to be thorough. The detail in describing the improvements possible to this work via both the EIC and SoLID data is very appreciated. This paper clearly meets the requirements to be published in this journal.

There is a general comment for the author. Many of the figures are very small and hard to read. It would be appreciated if figures could be re-arranged to be more visible. In this line of thinking, it would be a good idea to remove the inset g_T plot in Figs. 3 and 4. and have that plot stand on its own next to the individual flavor charge plots. Although I do understand that depending on how the plot is generated this could be a non-trivial task, and so could be skipped.

As a general question, it is mentioned that other future experiments, an upgraded RHIC, Belle II, COMPASS, will also provide useful data. Is there a particular reason that the focus here is on SoLID and EIC only?
  • validity: -
  • significance: -
  • originality: -
  • clarity: -
  • formatting: -
  • grammar: -

Author:  Daniel Pitonyak  on 2022-03-04  [id 2267]

(in reply to Report 1 on 2022-02-28)
Category:
answer to question

I thank the referee for their review of the manuscript and overall positive comments. In the updated version, I have tried to make the figures larger while also trying keep the length not too much over the limit. Regarding the gT inset, it is a nontrivial task to separate it out, so I did not do this, but hopefully with the larger figures, they are more readable. Concerning the question of why we focused on EIC and SoLID, it is more out of practicality. This paper was born out of work on the EIC Yellow Report and also colleagues at JLab requesting a similar study for SoLID. That is, those were the two facilities we had pseudo-data for. If other experiments in the future provide us with the same, then such an analysis can certainly be carried out for them. The choice to use EIC and SoLID was in no way to diminish the potential impact of other experiments.

---

## Editorial Decision

published